# Effects of a three-week executive control training on adaptation to task difficulty and emotional interference

**Rosa Grützmann**[1]*, **Norbert Kathmann**[1], **Stephan Heinzel**[2]

**1** Department of Psychology, Humboldt-Universität zu Berlin, Berlin, Germany, **2** Department of Education and Psychology, Freie Universität Berlin, Berlin, Germany

* gruetzmr@hu-berlin.de

**Data Availability Statement:** All data files are available from the Open Science Framework database (https://osf.io/t8z6h/).

## Abstract

Intact executive functions are characterized by flexible adaptation to task requirements, while these effects are reduced in internalizing disorders. Furthermore, as executive functions play an important role in emotion regulation, deficits in executive functions may contribute to symptom generation in psychological disorders through increased emotional interference. Thus, the present study investigated transfer effects of a three-week executive control training on adaptation to task difficulty and emotional interference in healthy participants ($n = 24$) to further explore the training's suitability for clinical application. To assess the adaptation to task difficulty, the proportion congruency effect on behavioral data (response times, error rates) and ERP measures (N2, CRN) was assessed in a flanker task with varying frequency of incompatible trials (25%, 75%). To quantify emotional interference, flanker stimuli were superimposed on neutral or negative pictures. Replicating previous results, the training increased interference control as indexed by decreased response times and errors rates, increased N2 amplitude and decreased CRN amplitude in incompatible trials after training. Proportion congruency effects were weaker than expected and not affected by the training intervention. The training lead to a shift in the time-point of emotional interference: before training negative pictures lead to a reduction in CRN amplitude, while after training this reduction was observed for the N2. This pattern illustrates that the training leads to a change in task processing mode from predominant response-related cognitive control to predominant stimulus-related cognitive control (N2), indicating a proactive processing mode.

## 1. Introduction

Executive functions are essential to intentional behavior and cognitive control. They support goal-directed behavior and adaptation to varying situational requirements [1–3]. Three core executive functions have been identified: inhibition/ interference control, working memory, and cognitive flexibility [1, 4]. Interference control or inhibition is defined as the ability to suppress prepotent stimuli (attentional inhibition) or automatically generated response tendencies

**Funding:** This work was supported by the German Research Foundation (DFG) grant GR 4901/2-1 and HE 7464/4-1. The funder did not play any role in the study design, data collection and analysis, decision to publish, or preparation of the manuscript.

**Competing interests:** The authors have declared that no competing interests exist.

(response inhibition). Working memory is defined as a brain system that provides temporary storage and manipulation of information [5]. Cognitive flexibility relies on the other two executive functions and comprises, amongst others, the ability to quickly and flexibly adapt to changing circumstances [1].

Mediofrontal-negativities in the event-related potential (ERP) of the EEG can serve as psychophysiological indicators of interference control. The N2 emerges 200–350 ms after stimulus presentation and is usually larger for incompatible trials, linking it to conflict processing [6–8]. The correct-related negativity (CRN) occurs 0–100 ms after correct responses and is thought to reflect response monitoring and strategy adaptation [9–11]. The error-related negativity (ERN) occurs 0–100 ms after erroneous responses. It has a larger amplitude than the CRN and is thought to reflect error detection and processing [12, 13]. As these components are all generated in the anterior cingulate cortex [8, 14, 15], they may reflect the same process, namely recruitment and implementation of cognitive control, activated at different time points and with varying intensity [10, 11, 16, 17].

As pointed out before, a core feature of intact executive functions is the ability to flexibly adapt to situational requirements (i.e. cognitive flexibility). Regarding interference control, this flexible adaptation is, amongst others, evident in the "proportion congruency effect". In conflict tasks, a higher proportion of conflict stimuli results in improved conflict resolution as reflected in decreased response time and enhanced accuracy for incompatible trials [10, 11, 16, 18–22]. Consequently, the response time difference between congruent and incongruent trials (i.e. the congruency effect) is reduced [23]. In-depth analysis of behavioral data by response time distributional analysis [24] indicates that the proportion congruency effect is caused by increased interference inhibition [10, 18, 19, 25]. On the ERP level, this is accompanied by an increase in conflict-related cognitive control, as reflected in the N2 amplitude, and a decrease in response-related cognitive control as reflected in the CRN amplitude in incompatible trials [10, 11, 16, 26].

Deficits in executive function are a common finding in psychological disorders [27–29]. These alterations also manifest on the ERP level. Overactive performance monitoring and interference control, as indexed by the N2, CRN and ERN have been found in internalizing disorders such as obsessive-compulsive disorder (OCD) and anxiety disorders [30–39], while externalizing disorders such as attention deficit hyperactivity disorder and substance use disorders are associated with reduced error monitoring [34–36]. As these alterations are also presented in unaffected first-degree relatives [34, 39] and prospectively predict symptom development [40–42], they most likely represent a risk factor and thus a promising target for interventions.

Besides general overactivity, reduced flexibility of interference control has also been observed in internalizing disorders. OCD appears to be associated with decreased flexibility as evident in an excessive activation of interference control and conservative response strategies in task conditions with low executive demand, resulting in failure to further increase interference control in more difficult conditions [18, 43, 44]. Similar patterns of impaired flexibility have also been observed for behavioral and neuronal correlates of working memory in OCD [45–49]. Furthermore, altered adaptation to conflict repetition has been found in OCD [30], generalized anxiety disorder [50], panic disorder [51] and high trait anxiety [52]. Against this background, interventions targeting the flexibility of executive functions may be especially beneficial for clinical applications.

Furthermore, interference control is crucial for the inhibition of maladaptive or irrelevant emotional information. Thus, it plays a vital role in emotion regulation [53, 54]. Emotional stimuli automatically capture attention [55, 56], are processed preferentially [57, 58] and can quickly trigger evolutionary-based behavioral tendencies. Hence, irrelevant emotional stimuli

can interfere with goal-directed intentional behavior, for example slowing response times or increasing error rates in experimental tasks (e.g. [59–64]). Brain networks associated with interference control are also activated during inhibition of emotional distraction [65, 66]. As illustrated in the dual mechanisms of control framework [67, 68], "cognitive" and "emotional" processes and brain networks show dynamic reciprocal interactions. Emotional distraction effects are stronger when activation of interference control is low, as for example on non-conflict trials [53, 54, 69] or in lower executive load [16].

Thus, by contributing to emotional dysregulation via reduced control over emotional interference caused by external (i.e. phobic objects) or internal stimuli (i.e. automatic maladaptive cognitions, intrusive thoughts), executive deficits may play a role in disorder development or maintenance. Interventions that increase interference control may also enhance inhibition of emotional distraction, thereby strengthening their clinical utility. First studies have shown that increases in executive control can reduce maladaptive affective processing as reflected in rumination [70] or interference by negatively-valenced emotional distractors [16, 53, 54].

Previous studies have shown that executive control training can successfully increase interference control [71–76], but it remains open whether this also leads to a) increased adaptation to task difficulty and b) reduced emotional interference. As deficits in these domains are often observed in psychological disorders, a transfer effect of a training intervention on these functions would strengthen its clinical utility.

In a previous study, we assessed the effects of an adaptive and a non-adaptive executive control training on primary indicators of cognitive control (N2, CRN and ERN amplitudes, responses times, error rates) and found superior effects of the adaptive training procedure [76]. Here, we extend on these findings and investigate near-transfer effects of the adaptive training procedure on adaptation to task difficulty and emotional interference. Specifically, a flanker task with two conflict frequency conditions (frequent compatible, FC: 25% incompatible trials, frequent incompatible, FI: 75% incompatible trials) was applied before and after the three-week adaptive executive control training and behavioral indicators (response times, error rate) and ERP correlates (N2, CRN) were assessed. We expected typical proportion congruency effects before training manifesting in a) shorter response times, b) smaller error rates, c) larger N2 amplitudes and d) smaller CRN amplitudes in incompatible trials in the FI than in the FC condition. If the training increases the flexibility of interference control, these proportion congruency effects should be larger after training than before training, i.e. differences scores between the FI and FC condition should be larger. To also assess the training's transfer effect on emotional interference control, negative and neutral pictures from the international affective picture system (IAPS [77]) were presented during the flanker task. Before training, we expected emotional interference effects as evident in a) prolonged response times, b) increased error rates, c) decreased N2 amplitudes and d) decreased CRN amplitudes after negative pictures. If the training successfully increases inhibition of emotional interference, these effects should be reduced or absent after the training.

## 2. Methods

### 2.1 Participants

Participants were 24 adults (17 women) with a mean age of 24 years (SD: 6.31, range 18–45 years). All participants received verbal and written explanation of the purpose and procedures of the study, gave their written informed consent in accordance to the ethical guidelines of the Declaration of Helsinki and received 10€ per hour for the experimental sessions and a bonus of 50€ for the at-home training or course credit for their participation. The study was approved by the ethics committee of the Humboldt-Universität zu Berlin. All participants had

normal or corrected-to-normal vision and reported no neurological diseases or history of head trauma. Initially, 29 participants were recruited. Two participants discontinued study participation after the pre-test. Data of three participants were discarded because of failure to comply with study instructions (two participants did not conduct the training as instructed, as indicated by a pattern of random button presses in the training data; one participant completed only ten training sessions).

Sixteen participants of the current sample were also included in our previous report which contrasted the effects of an adaptive and a non-adaptive training procedure [76]. Each participant completed the training only once and completed a standard flanker task (presented in [76]) and the modified flanker task (presented in the current manuscript) at pre- and post-training. Both studies were conducted as pilot studies that explored the effects of the training intervention in preparation for application in a larger sample of OCD patients in a clinical study (pre-registered in the German Register of Clinical Studies, DRKS00016174). To pilot effects of the flanker task with emotional interference, a 50% larger sample was recruited to account for the more complex setup of the modified flanker task and the respective statistical analyses in comparison to the standard flanker task. Decisions on sample sizes were made a priori and not based on preliminary data analyses.

## 2.2 Stimuli and procedures

**2.2.1 Experimental task.** A modified version of the flanker interference task ([78, 79], see Fig 1) was presented using Presentation (Neurobehavioral Systems, San Francisco, CA). Stimuli were displayed in white against a black background on a 19-inch computer monitor (refresh rate 100 Hz). At a viewing distance of 70 cm, the set of arrows was approximately 1.2˚ of visual angle wide and 1.2˚ of visual angle tall. Each trial was either compatible (target and flanker arrows pointed in the same direction) or incompatible (target and flanker arrows pointed in opposite directions). Participants were instructed to respond as quickly and accurately as possible to the direction of the central target arrow by pressing the corresponding key. The experiment consisted of two blocks with a different frequency of incompatible trials. One block contained 25% incompatible trials (frequent compatible condition, FC), whereas the other block contained 75% incompatible trials (frequent incompatible condition, FI).

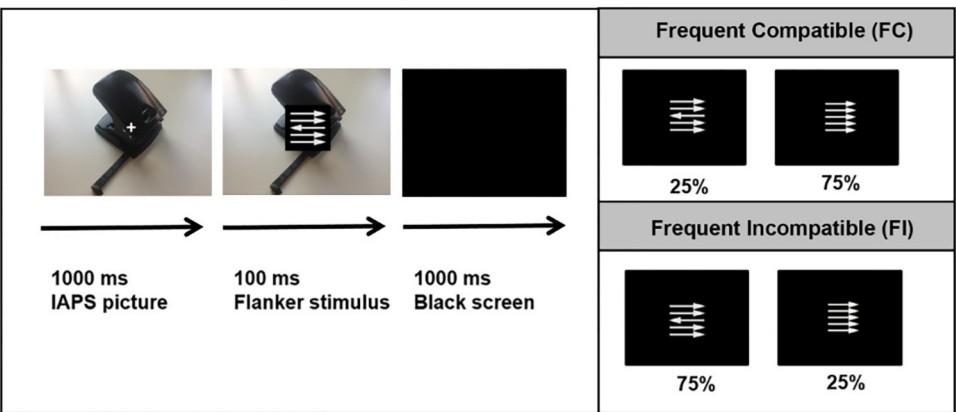

**Fig 1. Experimental design of the flanker task: Flanker stimuli were presented superimposed on neutral or negative IAPS pictures or OCD-related pictures.** Participants were instructed to respond fast and accurately with their left or right index finger to the target arrow. The experiment consisted of two conditions containing 25% (frequent compatible, FC) and 75% incompatible trials (frequent incompatible, FI condition). Because of copyright restrictions, the picture presented here is not from the IAPS but approximates the types of pictures presented during the experiment.

Block order was varied pseudo-randomly across participants. Target direction (left vs. right) was balanced within each block and stimulus direction was varied pseudo-randomly across trials. Each block comprised 288 trials, adding up to a total experiment duration of about 25 minutes. Starting 1000 ms prior to the flanker stimuli neutral ($M_{valence}$ = 5.24, $SD_{valence}$ = 0.42, $M_{arousal}$ = 3.11, $SD_{arousal}$ = 0.61) and negative pictures ($M_{valence}$ = 2.50, $SD_{valence}$ = 0.29, $M_{arousal}$ = 5.91, $SD_{arousal}$ = 0.73) from the International Affective Picture Viewing System (IAPS [77]) were presented. A list of the specific stimuli is presented in supplement 1 in S1 File. As the present experiment was part of a pilot study assessing the suitability for application in OCD, pictures related to OCD symptoms were also in one third of the trials presented (for further information please refer to supplement 2 in S1 File). As the current analysis focused on healthy participants, trials with OCD pictures were excluded, resulting in total number of 192 trials.

Then, flanker stimuli were presented superimposed on the IAPS pictures. Flanker stimuli and IAPS pictures were simultaneously turned off 100 ms after flanker onset, followed by a black screen for 1000 ms. Viewing angle for IAPS pictures was approximately 20.5˚ of visual angle wide and 15.4˚ of visual angle tall. Within each block and each stimulus type (compatible, incompatible) 50% were neutral and 50% were negative pictures. The same picture set (48 negative pictures, 48 neutral pictures) was used for the FI and the FC condition and for pre- and post-training sessions. Within each block each picture was presented two times. Picture order was varied pseudo-randomly across trials. In order to reduce eye movement artifacts, a white fixation cross was centrally superimposed on the IAPS pictures before flanker stimulus presentation. Participants were instructed to keep their gaze on the fixation cross during the presentation of the IAPS pictures.

The flanker task was applied before and after the three-week at-home executive control training. Preferably, pre- and post-test-sessions were scheduled to take place at the same day of the week on the same time exactly three weeks apart. If this was not possible, time between pre- and post-sessions was kept as close to three weeks as possible (mean number of days between pre- and post-session was 21 days, SD: 1.49, range: 20–23 days, mean time (in minutes) between pre- and post-session time of day was 48 minutes, SD: 7.11, range: 0–300).

**2.2.2 Executive control training.** After completing the pre-training test session, the participants were introduced to the training procedure. Participants were instructed to complete at least 15 training session within three weeks with no more than one session per day. As participants were allowed to conduct more than 15 sessions, a maximum of 21 training sessions could be completed within three weeks. Participants were included, if they had completed more than 10 training sessions. The mean number of completed sessions was 15 (SD: 1.09, Range: 12–16).

Each training session consisted of a flanker task training block and a n-back task training block, each lasting approximately 10–15 minutes (depending on individual response times and difficulty level) adding up to a total duration of 20–30 minutes. Task order was alternated every day. In the n-back task, numbers were presented consecutively on the screen. Targets were defined as reoccurrence of a number previously presented *n* trials before (i.e. one trial before in the 1-back condition, two trials before in the 2-back condition, etc.). In the 0-back condition, the target was defined as the number "0." The participants were instructed to press the space button with the thumb of their dominant hand when they recognized a target. In the flanker tasks compatible (target and flanker arrows pointed in the same direction) and incompatible (target and flanker arrows pointed in opposite directions) were presented and participants were instructed to respond as quickly and accurately as possible to the direction of the central arrow by pressing the left or right arrow key.

Participants started the training on the lowest difficulty level, which was increased step-wise from session to session, when participants' performance met the progression criterion (see

**Table 1. Difficulty levels in the flanker and nback task in the adaptive training procedure.** Interstimulusinterval (ISI) is reported in ms.

| Level | Flanker | | n-Back | |
| --- | --- | --- | --- | --- |
| | Load (Percentage of incompatible trials) | ISI | Load (n-Back-conditions) | ISI |
| 01 | 0%, 10%, 20% | 1600 | 0, 1, 2 | 1600 |
| 02 | | 1200 | | 1200 |
| 03 | | 800 | | 800 |
| 04 | 0%, 20%, 40% | 1600 | 0, 2, 3 | 1600 |
| 05 | | 1200 | | 1200 |
| 06 | | 800 | | 800 |
| 07 | 0%, 40%, 60% | 1600 | 0, 3, 4 | 1600 |
| 08 | | 1200 | | 1200 |
| 09 | | 800 | | 800 |
| 10 | 0%, 60%, 80% | 1600 | 0, 4, 5 | 1600 |
| 11 | | 1200 | | 1200 |
| 12 | | 800 | | 800 |

Table 1 for an overview of the difficulty levels). On each difficulty level, three different cognitive load conditions were presented. Variability of cognitive load conditions within each difficulty level was implemented, since deficits in executive functions are often not only characterized by reduced overall performance levels, but also by reduced cognitive flexibility. By applying variations within each difficulty level, we aimed to enhance flexible adaptation to environmental demands. Additionally, these variations might increase participants' motivation by reducing the task's monotony. The progression criterion for the flanker task was a correct response rate of at least 80% in incompatible trials in each cognitive load condition in the present session. The progression criterion for the n-back task was a target hit rate of at least 80% and a false alarm rate below 15% in each cognitive load condition in the present session. Difficulty was increased in a step-wise fashion by a combination of maximal cognitive load and time pressure. Cognitive load was increased by raising the amount of incompatible trials by 20% in the flanker task and by adding the next n-back condition in the n-back task [80, 81]. Time pressure was increased by shortening the inter-stimulus-interval by 400 ms. Difficulty levels were adapted independently for the two tasks with twelve levels within each task. If participants reached the maximum difficulty level before completing the training procedure, difficulty stayed on maximum for the remaining training sessions. Each flanker training session consisted of two easy blocks (0% incompatible trials) and four difficult blocks (amount of incompatible trials > = 10%). Each block comprised 100 trials. Each n-back session consisted of three easy blocks (0-back) and six difficult blocks (> = 1-back). Each block comprised 24 trials, containing six targets. Block order was pseudo-randomly varied within each session. Before each block, participants received information about the cognitive load level of the upcoming block ("easy block", "intermediate block", "difficult block").

Participants completed their training on their home computers. At the end of each session, a logfile containing response times, hit rates and difficulty level for both tasks was automatically transferred to the study coordinators. If the participants failed to complete at least five trainings sessions within one week, they were contacted by e-mail by the study coordinators and reminded to train regularly. Participants reached a mean difficulty level of 9 (SD: 3.45, range: 1–12) in the flanker task and a mean difficulty level of 8 (SD: 2.43, range: 4–12) in the n-back task.

## 2.3 Electrophysiological recording and data analysis

EEG and electroocculographic (EOG) activity were recorded continuously with 64 Ag-AgCl electrodes including Cz as recording reference. Electrodes were mounted with an EasyCap electrode system (Falk Minow Services, Munich, Germany) based on an equidistant electrode position system. Additional electrodes were placed on five external locations: IO1, IO2, Nz, neck and cheek. The electrode on the cheek served as ground. Electrode impedances were kept below 5 kΩ. The EEG was recorded with a sampling rate of 1000 Hz and a band pass filter of 0.01–250 Hz.

EEG data were processed with the Brain Vision Analyzer 2 (Brain Products, Munich, Germany). Eye-movement artifacts were corrected using an automatic ocular correction independent component analysis. Continuous EEG signals were filtered with a high-pass filter of 0.01 Hz and a low-pass filter of 30 Hz and re-referenced to average reference. For N2 analysis, stimulus-locked epochs with a duration of 800 ms including 200 ms pre-stimulus interval were extracted. The interval from -200 ms to 0 ms prior to the stimulus served as a baseline. For CRN analysis, response-locked epochs with a duration of 600 ms including 200 ms pre-response interval were extracted. The interval from -200 ms to 0 ms prior to the response served as a baseline. Epochs containing artifacts exceeding ± 200 μV in amplitude, voltage steps of more than 40 μV between consecutive data points or a minimal overall activity below 0.5 μV were excluded from further analysis. Flanker stimulus-locked averages for the N2 analysis and response-locked averages for the CRN analysis included only correct trials and were computed separately for each participant, for each test session (pre- and post-training), for compatible and incompatible trials, for each conflict frequency condition (FC, FI) and for each picture type (neutral, negative). Grand averages were filtered with a 15 Hz low-pass filter for visual presentation. ERPs were quantified as peak-to-peak amplitude, subtracting the preceding positive peak (time-window CRN: -100–25 ms pre-response, time-window N2: 150–350 ms post-stimulus) from the following negative peak (time-window CRN/ERN: 0–100 ms post-response, time-window N2: 200–400 ms post-stimulus). Peaks were set using an automatic procedure detecting the most negative/most positive value respectively at electrode FCz.

Statistical analyses were conducted with SPSS (Version 23.0, Chicago). Repeated-measurement analyses of variance (ANOVA) were used for statistical analyses of performance and ERP measures. Correct response times were analyzed using a 2 x 2 x 2 x 2 ANOVA including the factors time (pre-training, post-training), conflict frequency (FC, FI), compatibility (compatible, incompatible) and picture type (neutral, negative). Error response times and error rates in incompatible trials were analyzed using a 2 x 2 x 2 ANOVA including the factors time (pre-training, post-training), conflict frequency (FC, FI) and picture type (neutral, negative). N2 and CRN amplitudes were analyzed using a 3 x 2 x 2 x 2 x 2 ANOVA with the factors electrode (Fz, FCz, Cz), time (pre-training, post-training), conflict frequency (FC, FI), compatibility (compatible, incompatible) and picture type (neutral, negative). For all significant main effects or interactions, post-hoc comparisons were conducted using paired-samples t-tests. $T$-values, $p$-values, effect sizes (Cohen's $d$) and confidence intervals of effect sizes are reported for post hoc comparisons.

# 3. Results

## 3.1 Behavioral results

Error rates and response times are presented in Table 2 and in Fig 2.

The ANOVA on correct response times yielded a significant main effect of compatibility, $F(1,23) = 274.58$, $p < .001$, $\eta_p^2 = .92$, indicating shorter response times for compatible than for

**Table 2. Response times for correct responses in compatible and incompatible trials and error rates and error response times in incompatible trials after neutral and negative pictures in the FC and FI condition before and after training.** Error rates are presented in % and response times are presented in ms.

| | | Pre-Training | | | | | | | | Post-Training | | | | | | | |
| --- | --- | --- | --- | --- | --- | --- | --- | --- | --- | --- | --- | --- | --- | --- | --- | --- | --- |
| | | FC | | | | FI | | | | FC | | | | FI | | | |
| | | Neutral | | Negative | | Neutral | | Negative | | Neutral | | Negative | | Neutral | | Negative | |
| | | M | SD | M | SD | M | SD | M | SD | M | SD | M | SD | M | SD | M | SD |
| RT correct compatible | | 363.53 | 30.16 | 363.27 | 31.45 | 374.07 | 35.85 | 378.22 | 34.98 | 356.49 | 36.79 | 361.95 | 41.79 | 366.80 | 41.72 | 366.25 | 37.42 |
| RT correct incompatible | | 435.64 | 49.00 | 429.89 | 47.18 | 429.71 | 38.81 | 429.95 | 42.42 | 406.96 | 43.65 | 406.42 | 46.19 | 400.71 | 41.10 | 403.28 | 42.31 |
| RT error incompatible | | 349.58 | 35.64 | 361.36 | 30.62 | 364.03 | 42.23 | 366.71 | 29.99 | 372.10 | 87.33 | 350.33 | 29.50 | 367.72 | 42.84 | 373.51 | 40.91 |
| Error rate | | 19.10 | 15.53 | 23.50 | 13.83 | 8.67 | 6.22 | 10.06 | 5.39 | 14.43 | 12.88 | 14.93 | 10.09 | 6.95 | 6.10 | 8.67 | 7.32 |

incompatible trials. Additionally, a significant main effect of time was observed, $F(1,23) = 8.62$, $p = .007$, $\eta_p^2 = .27$, indicating shorter response times after training than before training. This effect was further specified by its interaction with compatibility, $F(1,23) = 31.98$, $p < .001$, $\eta_p^2 = .58$, indicating that response times in incompatible trials were significantly shorter after than before training, $t(23) = 3.23$, $p = .006$, $d = -0.79$, 95% CI [-1.51, -0.07], while no change

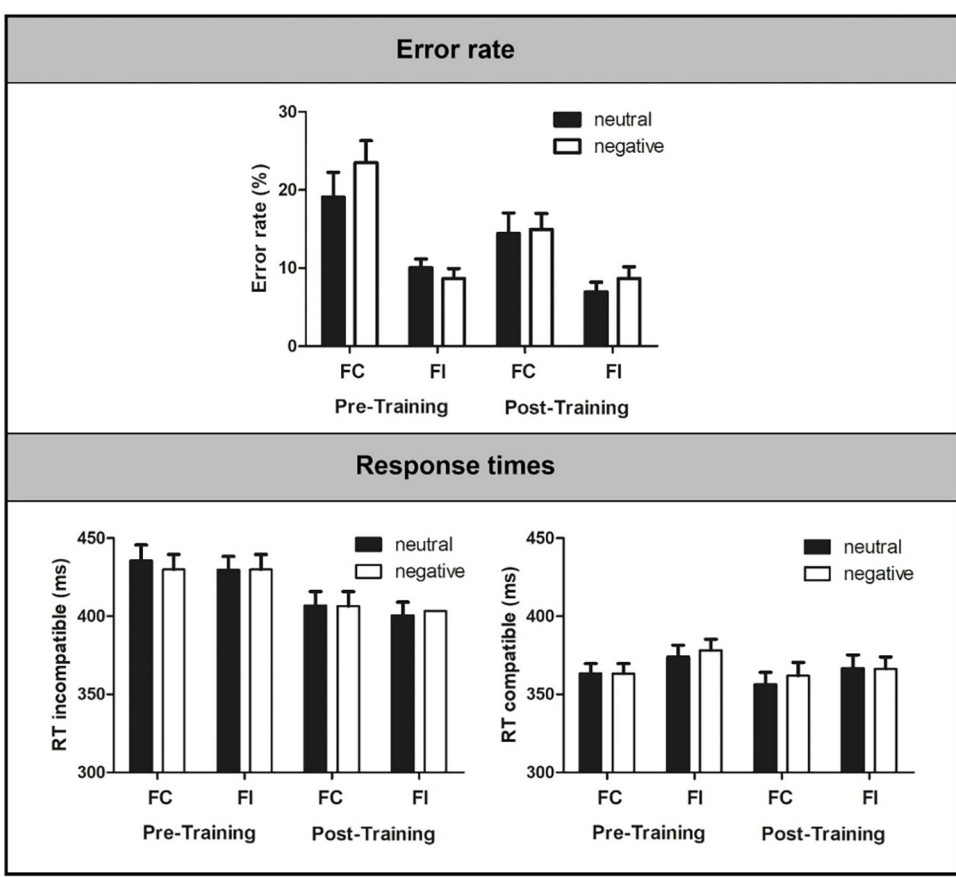

**Fig 2. Behavioral data (error rates in incompatible trials, response times in compatible and incompatible trials) in trials with negative and neutral pictures in the FC and FI condition before and after training.** Error rates are presented in %; response times are presented in ms. Bars represent standard errors. RT = response time, FC = frequent compatible, FI = frequent incompatible.

was observed for compatible trials, $t(23) = 0.96$, $p = .353$. The ANOVA on error response times did not yield any significant effects (all $F < 2.41$, all $p > .137$). The ANOVA on error rates yielded a significant effect of time, $F(1,23) = 6.49$, $p = .018$, $\eta_p^2 = .22$, indicating lower error rates after training than before training.

**3.1.1 Adaptation to task difficulty.** For correct response times, a significant interaction of executive load and compatibility was detected, $F(1,23) = 33.06$, $p < .001$, $\eta_p^2 = .59$. Post-hoc test showed that response times in compatible trials were significantly longer in the FI block, $t(23) = -3.73$, $p = .001$, $d = 0.77$, 95% CI [0.18, 1.35], while response times in incompatible trials did not differ between the FI and FC block. The ANOVA on response times yielded no evidence for an effect of the training on executive load modulations as none of the interactions involving the factors time and executive load reached significance (all $F < 0.61$, all $p > .443$).

The ANOVA on error rates yielded a significant main effect of executive load, $F(1,23) = 41.76$, $p < .001$, $\eta_p^2 = .65$. Error rates in incompatible trials were lower in the FI than in the FC block. Additionally, a significant interaction of time and executive load was observed, $F(1,23) = 5.43$ $p = .029$, $\eta_p^2 = .19$. Post-hoc test showed that error rate was significantly reduced after training in the FC block, $t(23) = 2.69$ $p = .013$, $d = -0.49$, 95% CI [-1.06, -0.09], while it remained stably low in the FI block, $t(23) = 1.29$, $p = .209$.

**3.1.2 Emotional interference.** The ANOVA on error rates yielded a significant effect of picture type, $F(1,23) = 4.43$, $p = .046$, $\eta_p^2 = .16$, indicating increased error rates after negative pictures. The ANOVA yielded no evidence for an effect of picture type on response times (all $F < 0.52$, all $p > .478$). Furthermore, the ANOVA yielded no evidence for an effect of the training on emotional interference on the behavioral level as none of the interactions involving the factors time and picture type reached significance (all $F < 2.47$, all $p > .130$).

## 3.2 N2 results

ERPs are depicted in Fig 3 and amplitude values are presented in Table 3. ERPs illustrating the proportion congruency effect are presented in the supplemental material S2 in S1 File.

The ANOVA yielded a significant main effect of electrode, $F(2,46) = 19.99$, $p < .001$, $\eta_p^2 = .47$, indicating that N2 was larger at Fz and FCz than at Cz (all $p < .001$), whereas amplitudes did not differ between Fz and FCz ($p = .320$). Furthermore, a significant interaction of electrode and compatibility was detected, $F(2,46) = 4.31$, $p = .045$, $\eta_p^2 = .16$, showing larger N2 amplitudes in incompatible trials than compatible trials at electrode Fz ($p = .003$) and FCz ($p = .064$), while no difference was detected at electrode Cz ($p = .793$).

A trend-level effect of time was detected, $F(1,23) = 3.87$, $p = .061$, $\eta_p^2 = .14$, indicating that the N2 was increased after training. This effect was further specified by its interaction with compatibility, $F(2,23) = 11.91$, $p = .002$, $\eta_p^2 = .34$, indicating that the N2 in incompatible trials was significantly larger after training, $t(23) = 3.10$, $p = .005$, $d = -0.74$, 95% CI [-1.32, -0.15], while no change was detected for compatible trials, $t(23) = 0.15$, $p = .882$.

**3.2.1 Adaptation to task difficulty.** A significant interaction of executive load and compatibility was observed, $F(1,23) = 16.76$, $p < .001$, $\eta_p^2 = .42$, indicating that the N2 in incompatible trials was significantly larger in the FC than in the FI condition, $t(23) = 4.23$, $p < .001$, $d = -0.88$, 95% CI [-1.47, -0.29], while this effect only reached a trend-level for the N2 in compatible trials, $t(23) = 1.90$, $p = .070$, $d = -0.39$, 95% CI [-0.96, 0.18]. The ANOVA yielded no evidence for an effect of the training on executive load modulations as none of the interactions involving the factors time and executive load reached significance (all $F < .09$, all $p > .361$).

**3.2.2 Emotional interference.** The ANOVA yielded several effects involving the factor picture type. A significant main effect of picture type was observed, $F(1,23) = 14.75$, $p = .001$, $\eta_p^2 = .39$, indicating reduced N2 amplitudes after negative pictures. The interaction of time

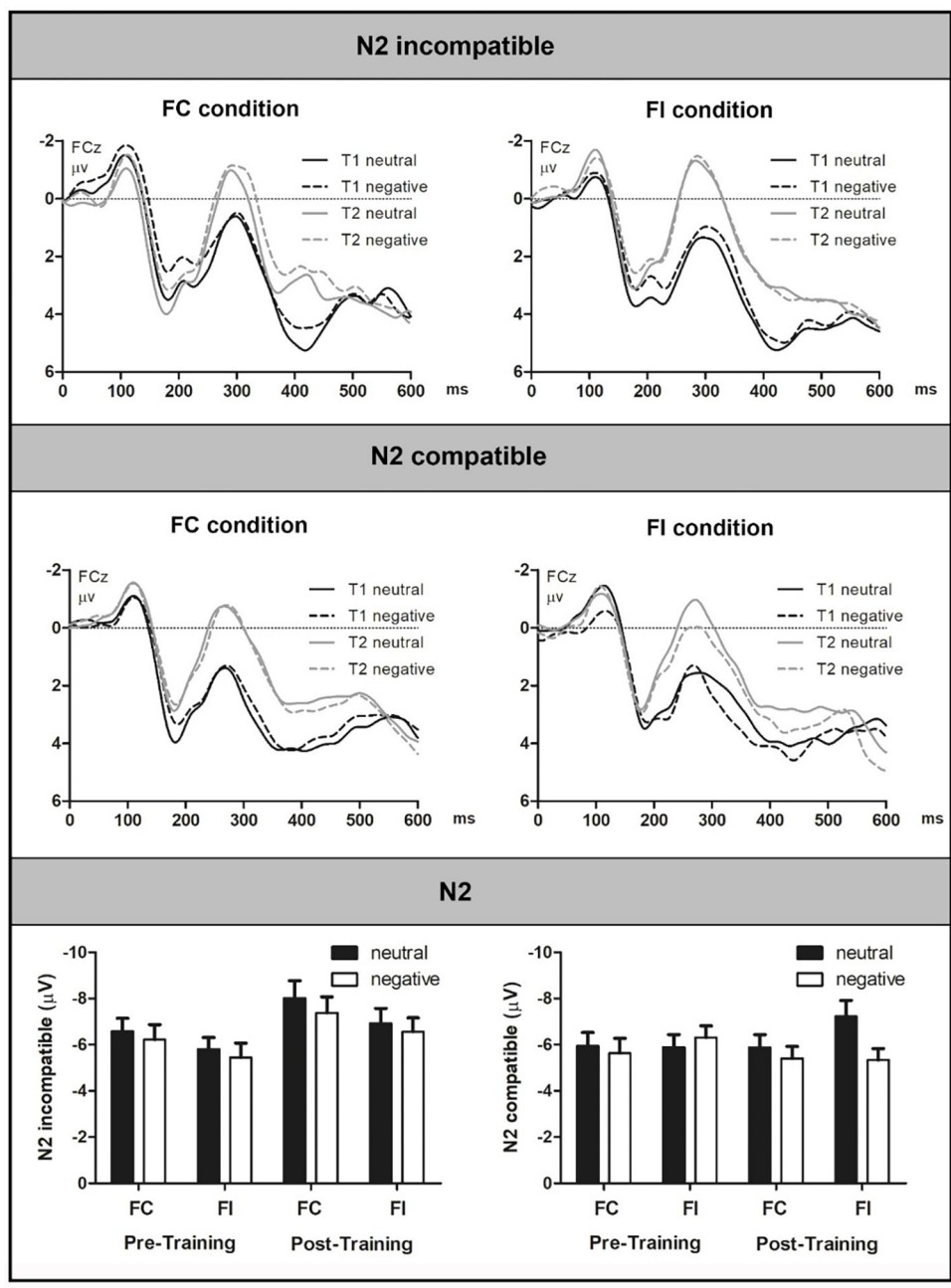

**Fig 3.** Grand Averages of the N2 in incompatible (upper panel) and compatible trials (middle panel) with neutral and negative pictures in the FC and FI condition at pre- (T1) and post-training (T2) at electrode FCz. The lower panel presents the CRN amplitude in each condition. Bars represent standard errors. FC = frequent compatible, FI = frequent incompatible.

and picture type, $F(1,23) = 6.63$, $p = .017$, $\eta_p^2 = .22$, showed that this reduction was significantly stronger after training, $t(23) = 4.44$, $p < .001$, $d = -0.84$, 95% CI [-1.43, -0.25], than before training, $t(23) = 0.84$, $p = .408$. This effect was further specified by the three-way interactions time * compatibility * picture type, $F(1,23) = 3.40$, $p = .078$, $\eta_p^2 = .13$, and time * executive load * picture type, $F(1,23) = 5.65$, $p = .026$, $\eta_p^2 = .20$, and their four-way interaction time * executive load * compatibility * picture type, $F(1,23) = 8.51$, $p = .008$, $\eta_p^2 = .27$. Generally, the

**Table 3. Amplitudes (in μV) of the N2 and CRN in compatible and incompatible trials after neutral and negative pictures in the FC and FI condition before and after training at electrode Fz, FCz and Cz.**

| | | Pre-Training | | | | | | | | Post-Training | | | | | | | |
|---|---|---|---|---|---|---|---|---|---|---|---|---|---|---|---|---|---|
| | | FC | | | | FI | | | | FC | | | | FI | | | |
| | | Neutral | | Negative | | Neutral | | Negative | | Neutral | | Negative | | Neutral | | Negative | |
| | | M | SD | M | SD | M | SD | M | SD | M | SD | M | SD | M | SD | M | SD |
| **N2 incom** | | | | | | | | | | | | | | | | | |
| | Fz | -5.77 | 3.59 | -5.48 | 3.42 | -6.12 | 3.49 | -5.16 | 3.54 | -8.09 | 3.87 | -7.08 | 3.33 | -7.19 | 3.47 | -6.89 | 3.29 |
| | FCz | -6.58 | 2.75 | -6.21 | 3.16 | -5.80 | 2.45 | -5.44 | 3.02 | -8.02 | 3.69 | -7.38 | 3.35 | -6.92 | 3.17 | -6.55 | 2.93 |
| | Cz | -4.12 | 2.79 | -3.79 | 3.16 | -2.91 | 2.53 | -2.94 | 2.77 | -4-97 | 3.63 | -4.61 | 2.93 | -4.27 | 3.05 | -3.61 | 2.71 |
| **N2 com** | | | | | | | | | | | | | | | | | |
| | Fz | -4.94 | 2.87 | -4.57 | 2.77 | -5.37 | 2.74 | -5.48 | 2.58 | -5.78 | 3.05 | -5.21 | 2.69 | -7.13 | 3.19 | -5.68 | 2.84 |
| | FCz | -5.95 | 2.81 | -5.63 | 3.12 | -5.89 | 2.62 | -6.30 | 2.52 | -5.87 | 2.74 | -5.40 | 2.55 | -7.22 | 3.38 | -5.33 | 2.45 |
| | Cz | -4.68 | 3.77 | -4.17 | 4.08 | -3.76 | 4.12 | -4.94 | 3.87 | -3.61 | 2.92 | -3.57 | 2.54 | -4.74 | 4.08 | -2.75 | 2.78 |
| **CRN incom** | | | | | | | | | | | | | | | | | |
| | Fz | -2.34 | 2.53 | -2.62 | 2.48 | -1.32 | 2.11 | -1.27 | 1.87 | -2.40 | 2.87 | -2.02 | 1.99 | -1.00 | 2.22 | -0.66 | 1.88 |
| | FCz | -4.13 | 2.65 | -3.41 | 2.83 | -2.84 | 2.48 | -2-22 | 2.63 | -2.79 | 3.32 | -2.24 | 2.21 | -1.69 | 2.59 | -1.82 | 2.18 |
| | Cz | -3.79 | 2.90 | -2.73 | 2.90 | -2.92 | 2.85 | -1.89 | 3.55 | -1.99 | 3.11 | -1.17 | 3.00 | -1.30 | 3.08 | -1.90 | 2.84 |
| **CRN com** | | | | | | | | | | | | | | | | | |
| | Fz | -0.88 | 2.99 | -1.07 | 2.70 | -1.86 | 3.48 | -2.31 | 3.19 | -1.20 | 2.32 | -1.05 | 2.92 | -1.64 | 2.55 | -2.04 | 2.22 |
| | FCz | -0.97 | 2.77 | -1.14 | 1.96 | -1.89 | 3.87 | -2.10 | 3.12 | -0.97 | 2.83 | -1.05 | 2.30 | -1.77 | 2.58 | -2.12 | 2.22 |
| | Cz | -0.76 | 2.54 | -0.72 | 2.10 | -1.20 | 3.73 | -0.90 | 3.19 | -0.01 | 2.89 | -0.36 | 2.26 | -0.84 | 2.55 | -1.50 | 2.69 |

reduction of the N2 amplitude after negative pictures after trainings was more pronounced in compatible trials (compatible: $t(23) = 3.74$, $p = .001$, $d = -0.86$, 95% CI [-1.45, -0.27], incompatible: $t(23) = 2.09$, $p = .048$, $d = -0.47$, 95% CI [-1.04, 0.11]) and in the FI condition (FI: $t(23) = 3.76$, $p = .001$, $d = -0.86$, 95% CI [-1.46, -0.27], FC: $t(23) = 2.10$, $p = .046$, $d = -0.45$, 95% CI [-1.02, 0.12]). Additionally, the reduction after negative pictures was stronger after training than before training. Consequently, the strongest modulation was observed in compatible trials in the FI condition after training, $t(23) = 3.73$, $p = .001$, $d = -0.70$, 95% CI [-1.28, -0.11].

### 3.3 CRN results

ERPs are depicted in Fig 4 and amplitude values are presented in Table 3. ERPs illustrating the proportion congruency effect are presented in S1 Fig in S1 File.

The ANOVA yielded a significant main effect of compatibility, $F(1,23) = 10.28$, $p = .024$, $\eta_p^2 = .31$, indicating that CRN amplitudes were larger in incompatible than in compatible trials. The interaction of electrode and compatibility, $F(2,46) = 8.45$, $p = .006$, $\eta_p^2 = .27$, indicated that this effect was present at electrode Cz ($p < .001$) and FCz ($p = .002$), but absent at electrode Fz ($p = .585$).

A significant interaction of compatibility and time was observed, $F(1,23) = 9.92$, $p = .004$, $\eta_p^2 = .30$. The CRN in incompatible trials was reduced after training, $t(23) = -2.36$, $p = .027$, $d = 0.53$, 95% CI [-0.08, 1.07], while the CRN in compatible trials remained stable, $t(23) = -0.30$, $p = .764$.

**3.3.1 Adaptation to task difficulty.** A significant interaction of compatibility and executive load was observed, $F(1,23) = 24.86$, $p < .001$, $\eta_p^2 = .52$, indicating that while the CRN in incompatible trials was reduced in the FI condition $t(23) = -3.82$, $p = .001$, $d = 0.78$, 95% CI [0.18, 1.36, the CRN in compatible trials increased, $t(23) = 2.83$, $p = .010$, $d = -0.66$, 95% CI [-1.24, -0.08]. Furthermore, a significant interaction of time, electrode and conflict frequency was detected, $F(1,46) = 4.34$, $p = .019$, $\eta_p^2 = .16$, indicating that the training led to a numerical

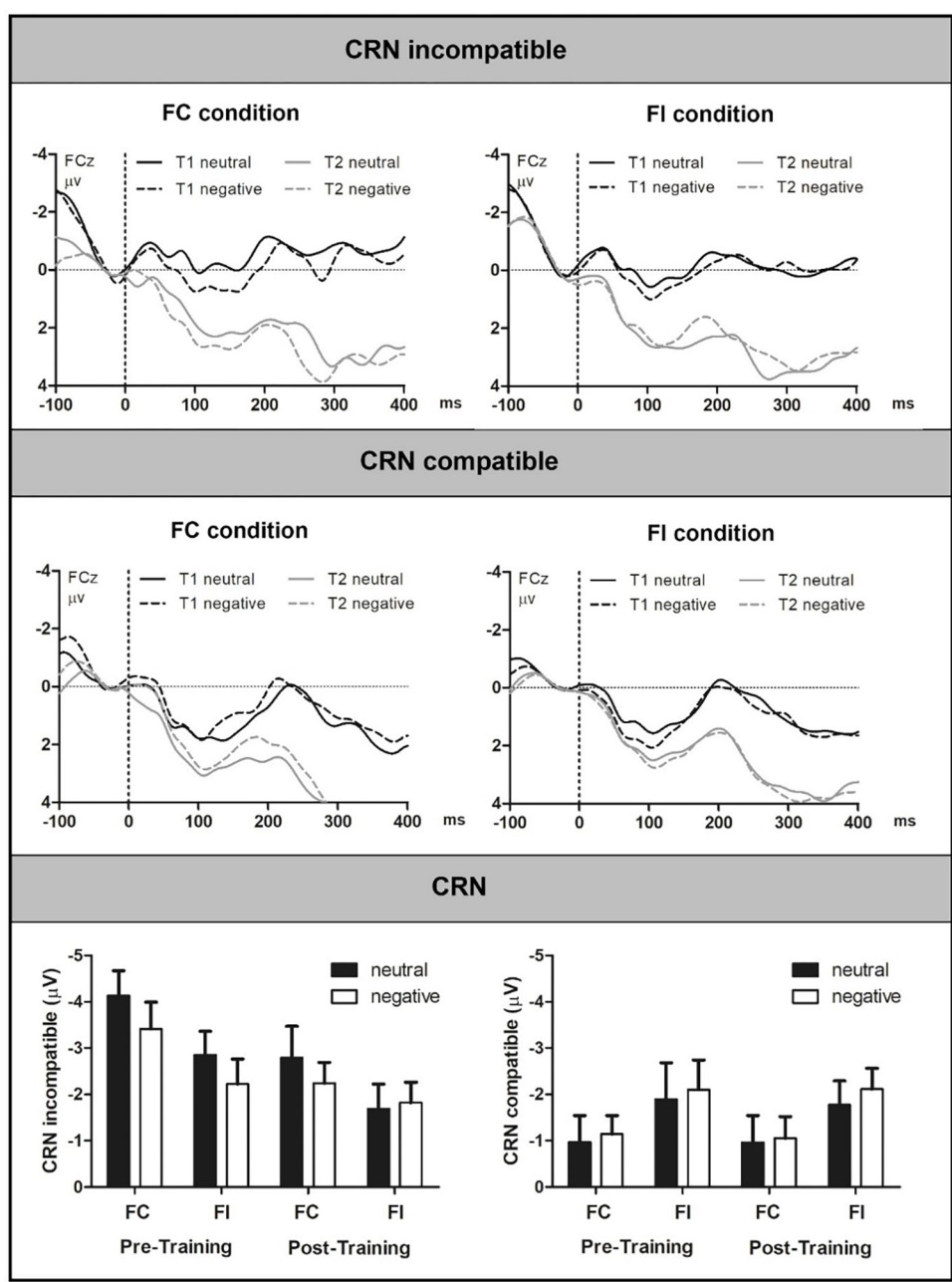

**Fig 4. Grand Averages of the CRN in incompatible (upper panel) and compatible trials (middle panel) with neutral and negative pictures in the FC and FI condition at pre- (T1) and post-training (T2) at electrode FCz.** The lower panel presents the N2 amplitude in each condition. Bars represent standard errors. FC = frequent compatible, FI = frequent incompatible.

increase of the block effect at electrode Cz. While amplitudes did not differ between the FC and FI before training, $t(23) = -0.75$, $p = .461$, a trend for larger CRN amplitudes in the FI condition was observed after training, $t(23) = 1.75$, $p = .093$, $d = -0.35$, 95% CI [-0.92, 0.22].

**3.3.2 Emotional interference.** A significant interaction of compatibility and picture type was observed, $F(1,23) = 9.92$, $p = .004$, $\eta_p^2 = .30$, indicating that CRN was trend-level reduced after negative pictures in incompatible trials, $t(23) = 1.84$, $p = .079$, $d = 0.36$, 95% CI [-0.21,

0.93], while no effect of picture type was detected in compatible trials, $t(23) = -1.19$, $p = .245$. Furthermore, a significant interaction of time, electrode and picture type was observed, $F(2,46) = 7.89$, $p = .007$, $\eta_p^2 = .26$. This interaction indicated that the reduction of the CRN after negative pictures was most pronounced at electrode Cz, where it was only present before training, $t(23) = 2.55$, $p = .018$, $d = 0.54$, 95% CI [-0.04, 1.11], while the difference between negative and neutral pictures was not significant after training, $t(23) = -1.01$, $p = .325$.

## 4. Discussion

In the present study, we investigated near-transfer effects of an adaptive executive control training to two domains, namely adaptation to task difficulty and inhibition of emotional interference. Several psychological disorders are associated with alterations in interference control. As these alterations may be further characterized by reduced adaptation to task requirements and impaired inhibition of emotional interference, executive training procedures targeting these domains might be especially promising for clinical application. We applied an adaptive three-week executive control training, that targets interference control (flanker task) and working memory components (n-back task) and has previously been shown to successfully enhance interference control on the behavioral and ERP level [76].

At baseline, a typical proportion congruency effect was observed for error rates, which were lower in the FI than in the FC condition, indicating increased interference control in the FI condition. Furthermore, a typical proportion congruency effect was observed for the CRN, with decreased amplitudes of the incompatible CRN and increased amplitudes of the compatible CRN in the FI condition. However, in contrast to previous studies, the modulation of response times and N2 amplitudes was more pronounced for compatible than for incompatible trials. For the N2 in incompatible trials, an unusual pattern of higher N2 amplitudes in the FC than in the FI condition was observed. Taken together, the pattern of proportion congruency effect was less consistent than in previous studies. It appears plausible, that this might be related to the unpredictable picture presentation. As picture valence was randomized across trials, participants could not anticipate which picture type would be presented on each trial. Unpredictable threat has high evolutionary relevance and leads to increased defensive activation, as indexed by increased fear-potentiated startle [82–84], and hypervigilance, as indexed by increased recruitment of attention networks [85–89]. Thus, the unpredictable threat induced by the random picture presentation may have resulted in altered activation of attentional processes, thereby obscuring typical proportion congruency adaptations. Furthermore, the current results illustrate that although the proportion congruency effect usually creates a complementary modulation of the N2 and CRN [10, 11, 16], and these opposing modulations have previously been found to be correlated on the inter- and intra-individual level [10], these modulations are not necessarily directly coupled and can also occur independently of each other as in the present study.

Regarding the primary training effect, results confirmed that the adaptive executive control training resulted in increased interference control as evident in reduced response times and reduced error rates after training. These behavioral effects were accompanied by increased N2 amplitudes and decreased CRN amplitudes, reflecting increased stimulus-locked interference control and decreased response-locked strategy adaptation. As previously described, these effects were specific to incompatible trials [76]. Please note that as the samples were overlapping and the modified flanker task contains elements of the standard flanker task analyzed in our previous report, this should not be valued as an independent replication. Still, these results illustrate, that the training successfully enhanced the targeted executive control functions.

Contrary to our expectation, the training did not produce near-transfer effects on adaptation to task difficulty as reflected in the proportion congruency effect. This is especially

surprising since the training directly incorporated elements aimed to enhance adaptation flexibility: Even though the *overall* difficulty progressively increased based on the participant's performance, *varying difficulty levels* were presented on each stage. Thus, participants were required to continuously adapt their behavior to changing task requirements. Additionally, participants were presented with information about the upcoming difficulty before each block ("easy block", "intermediate block", "difficult block"). This information was implemented to facilitate adaptation to task requirements by recruiting explicit, conscious adaptation mechanisms in addition to implicit mechanisms, thus possible strengthening training effects. Nevertheless, significant training-related changes in difficulty adaptation were not observed, neither for the behavioral nor for the ERP level. This lack of effect may be related to several factors. First of all, the current pilot study investigated a sample of healthy young adults without a diagnosis of psychological disorders. Previous studies have demonstrated that this group usually shows fast, flexible and efficient adaptation to task requirements [9–11, 16, 18–23]. As the training was designed to alleviate adaptation deficits in clinical populations, the lack of modulation may be due to a ceiling effect in the healthy study sample. Thus, the training may yield effects on difficulty adaptation in populations exhibiting pre-training deficits. Furthermore, the training incorporated explicit information about the upcoming difficulty before each block. This information was not presented in the transfer task before and after training. Consequently, the training may predominantly have enhanced conscious difficulty adaptation strategies that, due to the lack of cues, could not be applied in the transfer task.

However, the training modulated the susceptibility to emotional interference on the ERP level. Specifically, the training changed the time-point at which task processing is affected by emotional interference. For both ERP components a significant modulation of the emotional interference effect by the factor time was observed. For the CRN, emotional interference was stronger before training than after training. For the N2, the opposite pattern was observed. Here, emotional interference was stronger after training than before training. Interestingly, this mirrors the main effect of the training on these two components: After training, the N2 was increased, while the CRN was reduced. As both components share a similar topography it has been argued that they are generated by the same brain region and reflect similar functional processes [8, 10, 11, 14–17, 26]. In this line of thought, complementary modulation of these two ERP components may reflect a shift in the primary time-point of control application. Improved task processing after training appears to be characterized by enhanced stimulus-locked application of interference control, as reflected in the N2, and reduced response monitoring and reactive trial-to-trial strategy adaptation, as reflected in the CRN. The present data indicates, that emotional interference manifests in the predominant time-window of control application, which is shifted after training. Altered cognitive control in internalizing disorders mainly affects response-locked processes (i.e. the CRN and ERN) and appears to be strongly linked to emotional processes such as subjective error salience and threat sensitivity [90]. Thus, the training procedure may be applied to target these alterations by reducing both the overall magnitude of and the susceptibility to emotional influence of post-response control application.

It is important to note that, contradicting our expectations, the training did not result in a general reduction of emotional interference on the behavioral level. An increased error rate after negative pictures was observed before training, but this effect was not significantly modulated by the training. Still, the numerical values indicate a reduction of the emotional interference effect in the FC condition after training ($\Delta_{negative-neutral}$ before training: 4.39%, SD: 13.14, $\Delta_{negative-neutral}$ after training: 0.49%, SD: 7.77), but not in the FI condition $\Delta_{negative-neutral}$ before training: 1.39%, SD: 4.01, $\Delta_{negative-neutral}$ after training: 1.72%, SD: 4.42). This is in accordance with the dual mechanisms of control model [68] that postulates that interference is stronger

under conditions with low executive control and indicates that after training interference control is generally increased irrespective of conflict frequency condition. However, the according interaction in the ANOVA did not reach significance ($p$ = .130), possibly due to the large variability of error rates in the FC condition, the factorial complexity of the ANOVA and limited statistical power. Additionally, in accordance with previous studies in healthy participants, emotional interference effects at baseline were rather smaller [16]. Thus, possible training effects might again be obscured by ceiling effects and should be further explored in clinical population exhibiting stronger baseline deficits.

Some limitations need to be considered. The current design did not comprise a control condition, thus the observed changes may be partly caused by unspecific mechanisms. However, in a previous study [76], the adaptive training program was shown to create superior effects on interference control compared to a control condition (non-adaptive training) and the current study investigated the transfer effects of the previously established procedure. Due to practical reasons, both transfer effects (difficulty adaptation, emotional interference) were assessed in a single task. As the current data indicates that picture presentation may have reduced difficulty adaptation, transfer effects should preferably be assessed separately in future studies. Furthermore, the current design did not include trials without pictures. Thus, the interference effect created by general picture presentation irrespective of valence cannot be quantified. Finally, as participants were recruited from psychology students, they were comparably young and the majority was female. Thus, the generalizability of the findings may be limited. Since women have been shown to react more strongly to IAPS pictures [91–93], the effects might be smaller in male participants.

The current study replicates in a partially overlapping sample that a three-week executive control training increases interference control as reflected in decreased response times and error rates, decreased CRN and increased N2 amplitudes in incompatible trials. Near transfer effects to difficulty adaptation were not observed, but might have been limited by a ceiling effect in the healthy study sample. The training modulated the time-point of emotional interference: Before training, emotional interference mainly affected response-locked control processes (CRN), after training it manifested on stimulus-locked interference control (N2). These effects illustrate, that improved behavioral performance after training is accompanied by a change in processing mode in which response-locked processes lose importance in favor of stimulus-locked processes. As alterations of cognitive control in internalizing disorders mainly manifest in an increased magnitude of and increased susceptibility to emotional processes of response-locked control (CRN, ERN), the current training procedure may be helpful in alleviating these deficits by shifting the primary time-point of control application to the stimulus-locked time-window.

## Supporting information

**S1 File.**
(DOCX)

## Acknowledgments

### Author notes

The authors thank Rainer Kniesche, Thomas Pinkpank and Ulrike Bunzenthal for technical assistance. The authors also thank Marie Bartossek, Franziska Jüres, Jacqueline Kimm, Marlene Reissing, Janika Wolter-Weging and Maria Zadorozhnaya for assistance in data collection.

## Author Contributions

**Conceptualization:** Rosa Grützmann, Norbert Kathmann, Stephan Heinzel.

**Data curation:** Rosa Grützmann.

**Formal analysis:** Rosa Grützmann.

**Funding acquisition:** Rosa Grützmann, Stephan Heinzel.

**Investigation:** Rosa Grützmann.

**Methodology:** Rosa Grützmann.

**Project administration:** Rosa Grützmann, Stephan Heinzel.

**Resources:** Norbert Kathmann.

**Supervision:** Norbert Kathmann.

**Writing – original draft:** Rosa Grützmann.

**Writing – review & editing:** Rosa Grützmann, Norbert Kathmann, Stephan Heinzel.

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
