## [Decision Letter · Decision Letter 0]

19 Oct 2022

Effects of a three-week executive control training on adaptation to task difficulty and emotional interference

PONE-D-22-19876

Dear Dr. Grützmann,

We’re pleased to inform you that your manuscript has been judged scientifically suitable for publication and will be formally accepted for publication once it meets all outstanding technical requirements.

Kind regards,

Wi Hoon Jung, PhD

Academic Editor

PLOS ONE

1. Please include captions for your Supporting Information files at the end of your manuscript, and update any in-text citations to match accordingly. Please see our Supporting Information guidelines for more information: http://journals.plos.org/plosone/s/supporting-information.

Reviewers' comments:

Reviewer's Responses to Questions

**Comments to the Author**

1. Is the manuscript technically sound, and do the data support the conclusions?

Reviewer #1: Yes

Reviewer #2: Yes

2. Has the statistical analysis been performed appropriately and rigorously? 

Reviewer #1: Yes

Reviewer #2: Yes

3. Have the authors made all data underlying the findings in their manuscript fully available?

Reviewer #1: Yes

Reviewer #2: Yes

4. Is the manuscript presented in an intelligible fashion and written in standard English?

Reviewer #1: Yes

Reviewer #2: Yes

5. Review Comments to the Author

Reviewer #1: This work evaluated the effect of cognitive control training on adaptation to an increased difficulty of a cognitive task and the superimposed effect of inhibition of emotional interference. It is anticipated that the data generated may guide the development of a therapeutic exercise to treat patients with certain internalizing disorders (e.g., OCD).

The work has a direct antecedent in which the authors had already shown the supremacy of adaptive vs. non-adaptive executive control training concerning better executive control. Hence, the present study represents a logical extension of the original work.

The paper is very well written. The Introduction and Background sections provide helpful information for the readers. The methods and Results are sound. Discussion is well organized/structured.

Their evaluation of the cognitive control training in a group of 24 healthy young volunteers confirmed many expected results, mainly that the adaptive executive training resulted in increased interference control. Particularly exciting was the observation that the behavioral effects were accompanied by changes in the ERPs analyzed (i.e., increased N2 amplitude and decreased CRN amplitudes), interpreted as a transition in the processing mode: from a response-related to a stimulus-related cognitive control.

In conclusion, this is an exciting study showing promising results that should be tested in a clinical trial in the near future; therefore, I highly recommend that it be published without significant changes or alterations.

Reviewer #2: Authors performed rigorous research with appropriate statistical analysis.

The article is clearly written while addressing the research question coherently.

I recommend editors to accept this article for publication.

6. PLOS authors have the option to publish the peer review history of their article (what does this mean?). If published, this will include your full peer review and any attached files.

Reviewer #1: No

Reviewer #2: No

---

## [Editor Report · Acceptance letter]

11 Nov 2022

PONE-D-22-19876 

Effects of a three-week executive control training on adaptation to task difficulty and emotional interference 

Dear Dr. Grützmann:

I'm pleased to inform you that your manuscript has been deemed suitable for publication in PLOS ONE. Congratulations! Your manuscript is now with our production department. 

Kind regards, 

on behalf of

Dr. Wi Hoon Jung 

Academic Editor

PLOS ONE